# Green Facile Synthesis of Silver-Doped Zinc Oxide Nanoparticles and Evaluation of Their Effect on Drug Release

**DOI:** 10.3390/ma15165536

**Published:** 2022-08-11

**Authors:** Nadia Mahmoudi Khatir, Farzaneh Sabbagh

**Affiliations:** 1Department of Biotechnology, Faculty of Biological Sciences, Alzahra University, Tehran 1993891176, Iran; 2Department of Chemical Engineering, Chungbuk National University, Cheongju 28644, Korea

**Keywords:** Ag-ZnO nanoparticles, *κ*-Carrageenan, green synthesis, biopolymer, drug release

## Abstract

Silver doped zinc oxide nanoparticles (ZANPs) were synthesized by the gelatin mediated and polymerized sol-gel method, and a calcination temperature of 700 °C was applied for 2 h. X-ray diffraction (XRD), FESEM, TGA, DSC, and EDS were performed to study the structure of the prepared nano-powders. Both cubic silver and hexagonal ZnO diffraction peaks were detected in the XRD patterns. The XRD results, analyzed by the size strain plot (SSP) and Scherrer methods, showed that the crystalline sizes of these nanoparticles increased as the Ag concentration increased. The results were observed via transition electron microscopy (TEM), where the particle size of the prepared samples was increased in the presence of silver. Catechin was chosen as a drug model and was loaded into the hydrogels for release studies. The drug content percentage of catechin in the hydrogels showed a high loading of the drug, and the highest rate was 98.59 ± 2.11%, which was attributed to the Zn_0.97_Ag_0.03_O hydrogels. The swelling of the samples and in vitro release studies were performed. The results showed that Zn_0.91_Ag_0.09_O showed the highest swelling ratio (68 ± 3.40%) and, consequently, the highest release (84 ± 2.18%) within 300 min. The higher amount of silver ions in the hydrogel structure causes it to enhance the osmotic pressure of the inner structure and increases the relaxation of the structure chain.

## 1. Introduction

Nowadays, drug delivery systems based on nanoparticles have been used in a wide variety of applications. Zinc oxide (ZnO) nanoparticles with high stability and biocompatibility, excellent biomedical properties, and high selectivity are some of the best candidates for drug delivery systems [1,2]. Zinc oxide (ZnO) is an interesting multifunctional nanomaterial [3,4,5] that has established potential applications in various medical, biological [6,7,8,9], electronic, electrochemical, and magnetic fields, and their optical properties have attracted considerable attention from scholars [10,11,12]. Furthermore, its application is remarkable for resistive switching memory (RRAM) [13]. Additionally, ZnO nanoparticles can be used in ultraviolet (UV) photodetectors as controlling units [14], inflame detectors, and can identify different gases in nanosensors. The excitation binding energy of ZnO nanoparticles is approximately at room temperature, and an excitonic transition is possible, leading to a lower required voltage for laser emission. By changing the NPS nature of its dopants and morphology, the electronic and optical features of ZnO can be modified. When Ag is doped with ZnO, the size and morphology of ZnO NPs change significantly [15,16,17]. Silver-doped zinc oxide nanoparticles via green facile were investigated in some research [18,19]. Some features make ZnO additionally attractive for various applications, such as its biocompatibility, nontoxicity, cost-effectiveness [20,21,22,23], and saturation velocity of 3.2 × 10^7^ cm s^−1^ [24,25]. Multiple methods are available to prepare ZnO nanoparticles, such as microwave [26,27,28,29], reserve micelle [30], hydrothermal [31,32,33,34], chemical vapor deposition (CVD) [35], pyrolysis [36,37], DC thermal plasma synthesis [38], precipitation [39,40,41,42,43], combustion synthesis [44,45,46,47], and sol-gel [48,49,50,51,52,53]. Overall, its electronic performance is dependent on its size, shape, contamination, doping, and composition.

Moreover, the sizes of its crystallite and strain relaxation dynamics are concurrently significant on ZnO’s nanostructure features. Notably, the electron structures of ZnO nanoparticles are affected by the presence of defects and their quantum size, as well as dopants [36,37,38,39,40,41,42,43,44]. Subsequently, the ionic radii of the dopants differ from their zinc host. The variations of crystallite size alongside the lattice parameters are also important. In general, the quantity of the mediated defects in the lattice strain and crystallite size variations causes a broadening in diffraction peaks. Furthermore, a lattice mismatch or deficiency of dopants produces a lattice strain spreading along the lattice. Therefore, the lattice strain leads to peak positions via electrophonon coupling and intensity changes. The non-uniform strain causes the peak to widen, and the uniform strain changes the peak’s position [54]. Catechin is a flavonoid compound found in a variety of foods and herbs, including tea, grapes, apples, cacaos, and berries [55,56]. Furthermore, catechin is important in human health and has antioxidant properties [56,57]. A hydrogel is a polymer network that holds a large amount of water and has a semi-solid form [51,58]. Hydrogels are hydrophilic, porous, and have a soft network and are a crucial class of substances widely associated with bioanalytical chemistry, drug delivery, and soft robotics [59].

The aim of this study was to apply the doped nanoparticles of Zn_1−x_Ag_x_O into the κ-Carrageenan hydrogel and study the impact of doped nanoparticles on the release of catechin. In this study, we prepared Zn_1−x_Ag_x_O nanoparticles (ZA-NPs) at room temperature with the use of the sol-gel method in the gelatin media. We applied gelatin throughout the process of calcination, which is used as a polymerization and stabilizer agent to prevent the NPs’ growth. These ZA-NPs are fabricated by Ag doping to determine the variations in their crystal structures, size, and morphology and investigation about the texture, swelling, and drug delivery systems.

## 2. Materials and Methods

### 2.1. Materials

The zinc nitrate hexahydrate, sodium carboxymethyl cellulose (NaCMC), *κ*-Carrageenan, calcium carbonate (CaCO_3_), (+)-catechin hydrate, and gelatin ((NHCOCH–R1)n, R1 = amino acid, Type A, Porsin) that were utilized in this study were bought from Sigma-Aldrich (Tehran, Iran). All these substances were implemented as obtained with no extra purification. Distilled water was used for hydrogel synthesis. Zinc nitrate tetrahydrate (Zn(NO_3_)_2_._4_H_2_O) in analytical grade, silver nitrate (AgNO_3_), and distilled water were used to prepare the ZA-NPs in the form of Zn_1−x_Ag_x_O.

### 2.2. Green Synthesis of Nanoparticles

Chemicals (Zn(NO_3_)_2_._4_H_2_O) 6.425, 6.136, 5.856, 5.584 gr and AgNO_3_ 0, 0.123, 0.242, 0.358 gr were implemented, respectively, to obtain 2 g of the ultimate ZA-NP blend in the form of Zn_1−x_Ag_x_O. To produce ZA-NPs, gelatin with a ratio of (2:1) to the final product was added to 60 mL of distilled water, bit by bit. Additionally, (Zn(NO_3_)_2_._4_H_2_O) and AgNO_3_ were liquefied into 20 mL of distilled water separately. Then, we added both of them to the gelatin liquid. This liquid was constantly mixed in an oil bath at 85 °C. By dissolving the gelatin in the water, a clear liquid was obtained. Next, the Zn^2+^ and Ag^2+^ liquid was added to the obtained gelatin liquid. A viscous and transparent gel turned into an acquired one because of constantly stirring the solution at 85 °C for 6 h. Finally, the resulting gel was rubbed for the calcination process on the inner wall of a small amount of alumina crucible and put into a furnace at 700 °C. The rate of heating was 5 °C/min, and the duration of the heating time was 2 h [60]. Figure 1 shows the procedure of the nanoparticles’ synthesis.

### 2.3. Hydrogel Production

The *κ*-Carrageenan hydrogels were prepared using the method from [51]. Briefly, 0.10 g of CMC and 0.50 g of *κ*-Carrageenan were dissolved in distilled water (20 mL) at 80 °C. To prepare the hydrogel, 0.15 g CaCO_3_ was liquified in distilled water (10 mL), one by one. The solutions were mixed for 1 h until a clear, homogenous, and viscous solution with no bubble was achieved to fabricate the hydrogel. Finally, an aqueous solution of (ZnO, Zn_0.97_Ag_0.03_O, Zn_0.94_Ag_0.06_O, and Zn_0.91_Ag_0_._09_O) nanoparticles (0.8 mg/mL) were prepared according to [51] and was included to the polymer solution and mixed at ambient temperature. The ZnO sample was considered as the control in the hydrogels.

#### 2.3.1. Swelling of Hydrogels

The swelling ratio was calculated for a specific amount of hydrogels at ambient temperature and a pH of 8.5 using Equation (1):Swelling ratio% = W0 − Ws/W0 × 100(1)
where W0 and Ws are the original weight and the swollen weight of the hydrogel at the time, t, correspondingly.

#### 2.3.2. Catechin Loading

To encapsulate the catechin in the hydrogels, the prepared hydrogels were soaked in the catechin solution (5 mg/mL) at 4 °C for 2 days. After the absorption of catechin by the *κ*-Carrageenan hydrogels, the hydrogels were immersed in distilled water for around 5 min [58]. This step aimed to reduce the possibility of eruption in the release experimentations and to remove the extra adsorbed catechin on the surface.

#### 2.3.3. Characterization of Nanoparticles

XRD analysis was performed using a powder diffractometer (PHILIPS, PW1730, Amsterdam, The Netherlands) using Cu Kα radiation (λ = 1.54056 Å) with a voltage of 40 v and A = 30 mA (2θ = 20~80°, rate of 5 deg/min). The morphology of the nanoparticles was estimated via transmission electron microscopy (TEM) (Model CM120, Amsterdam, The Netherlands), with a maximum voltage of 100 kv. The chemical bonds of the complexes were analyzed using an FTIR (Bio Surplus, Thermo Nicolet Avatar 370, San Diego, CA, USA) at the wavelength of 4000–400 cm^−1^. To determine the adhesion, hardness, consistency, and springiness of the hydrogels, a texture analyzer (TA. XT. Plus, Stable Micro Systems, Godalming, UK), field emission scanning electron microscopy (FESEM) (MIRA III, TESCAN, Brno, Czech), and thermal stability, investigated via TGA Thermogravimetric Analysis (TGA) (TA, Q600, Shicago, IL, USA), were employed. The analytical probe P/0.4 (4 mm diameter, stainless steel cylinder) was compressed in the hydrogels with a trigger force of 3 g triplicate at a speed of 1.0 mm/s and 20 mm in height to a 60% height of the sample’s deepness.

#### 2.3.4. Drug Content

A dried hydrogel (2.0 g) was transferred in 1L of a phosphate buffer solution with a pH of 7.4 (USP 24) and stirred for 24 h at room temperature. Samples of 1 mL were withdrawn, diluted into 25.0 mL of the buffer, and centrifuged for 5 min at 12,800 rpm (Spectrafuge 24D, Labnet International, Inc., Edison, NJ, USA), and the supernatant was analyzed spectrophotometrically. The catechin concentration was determined by using a UV-spectrophotometer (Hitachi U-2001,Tokyo, Japan) at the wavelength of 317 nm. The real value of the drug content was calculated as the detected amount of the drug with respect to the theoretical amount of the drug. The catechin, expressed as a percentage, was determined in triplicate for all batches.

#### 2.3.5. In Vitro Release Studies

The provided samples were immersed in beakers full of 15 mL of PBS medium liquid (pH 8.5) and placed in an incubator at 50 rpm and 37 °C. Every single 60 min, a 4.0 mL aliquot of the solution was taken out, and a 4.0 mL of a new buffer was substituted to maintain the original volume after every single withdrawal. The concentration of the catechin released was determined using a UV–vis spectrophotometer. The amount of catechin was measured by interpolation from the catechin standard curve at l = 317 nm.

## 3. Results

### 3.1. XRD Characterization

Figure 2 presents the XRD designs of the prepared ZnO-NPs and ZA-NPs. The results show that the ZnO hexagonal structure is formed at the selected calcination temperature of 700 °C, and there is no more diffraction peak related to the impurity or other crystalline structures, as shown in Figure 2A. Figure 3 shows the XRD patterns of *κ*-Carrageenan/Zn_1–x_Ag_x_O/NaCMC/catechin; x = 0.0, 0.03, 0.06, and 0.09, but some more diffractions peaks were observed in the designs, as shown in Figure 2B–D, as Ag was put into the mixture. The appeared peaks at 2θ = 38.28°, 44.44°, and 64.57° are indexed to the cubic silver crystalline structure. The Bragg equation (λ=2dsinθ) was used to calculate the lattice parameters wherever d is the space between two miller indices (*hkl* panels) and *a*, *b*, and *c* are the lattice constants. The φ is the angle between the planes.

As shown in Table 1, the other parameters were obtained from the lattice geometry equation. It is for hexagonal structure [50,61] by:(2)$$\frac{1}{d^{2}}=\frac{4}{3}\left(\frac{h^{2}+hk+k^{2}}{a^{2}}\right)+\frac{l^{2}}{c^{2}}$$
(3)$$V=\frac{\sqrt{3}a^{2}c}{2}=0.866a^{2}c$$
(4)$$\cos\varphi =\frac{h_{1}h_{2}+k_{1}k_{2}+\frac{1}{2}\left(h_{1}k_{2}+h_{2}k_{1}\right)+\frac{3a^{2}}{4c^{2}}l_{1}l_{2}}{\sqrt{\left(h_{1}^{2}+k_{1}^{2}+h_{1}k_{1}+\frac{3a^{2}}{4c^{2}}l_{1}^{2}\right)\left(h_{2}^{2}+k_{2}^{2}+h_{2}k_{2}+\frac{3a^{2}}{4c^{2}}l_{2}^{2}\right)}}$$

It is impossible that the Ag atoms defuse in the hexagonal forms of ZnO-NPs and form as Ag nanostructures at the selected calcination temperature. It was found that some metals, such as silver, gold, and platinum, were highly stable and resisted the oxidation process. Therefore, they need to be heated over 1000 °C to be diffused into the ZnO hexagonal structure [62,63].

Figure 4 shows the two diffraction peaks related to the ZA-NPs. Figure 4A,B is related to the ZnO-NPs’ hexagonal structure and cubic Ag nanocrystals, respectively. As shown in Figure 4A, the intensity of the diffraction peaks decreased as the Ag atoms in the compound increased. This could be due to the transpired defects in the hexagonal structure by increasing the Ag impurity. As expected, the intensity of the curving peak in Figure 4B enhanced as the Ag amount increased in the mixture. The maximum intensity decreases of the hexagonal diffraction peak were observed for x = 0.09, and then the intensity was almost constant for the further amount of x in the ZA-NPs compounds.

There are some methods used for crystalline size calculations, such as the Scherrer, Williamson–Hall (W–H), and size–strain plot (SSP). The W–H and SSP techniques are more powerful and common techniques than the Scherrer technique, as the strain of the lattice affecting the position and intensity of the diffraction peaks is evaluated for the estimates. Between the W–H and SSP methods, the results obtained by an SSP are more reliable since the Gaussian function is used to analyze the diffraction peaks; therefore, more accurate results with fewer errors are obtained by the SSP method (Figure 5 and Figure 6). The XRD data achieved from this study were analyzed by the SSP technique. This technique, ${\left(d_{hkl}\beta_{hkl}\cos\theta \right)}^{2}$ is plotted regarding $\left(d_{hkl}^{2}\beta_{hkl}\cos\theta \right)$ using the following relation:(5)$${\left(d_{hkl}\beta_{hkl}\cos\theta \right)}^{2}=\frac{A}{D}\left(d_{hkl}^{2}\beta_{hkl}\cos\theta \right)+{\left(\frac{\varepsilon }{2}\right)}^{2}$$
where β at half the minimum intensity is the full peak width (FWHM), *A* is a constant and the plane distance equal to ¾, *θ* is the peak position, ε is the lattice strain, and *D* is the crystalline size. The slope of the fitted data presents the crystalline size. Figure 5 presents the plots, and Table 1 and Table 2 show the calculated results. It is indicated that the crystalline size of the ZA-NPs is not changed by Ag for x = 0 to 0.09, whereas the crystalline size increase is observed for the further amount of the Ag concentrations above x = 0.06.

It has been reported that impurities make some defects in the lattice that can decrease the crystalline size and terminate the crystal growth. Furthermore, it was observed that silver and gold were used as catalyzed to improve the growth of ZnO nanostructures [64,65]. Therefore, it can be concluded that x = 0 to 0.09 in the Zn_1−x_Ag_x_O compound. Both properties of the added silver atoms discussed above cover each other. For this reason, the crystalline sizes are the same for x = 0 to 0.09, whereas the Ag amount increases. In addition, the SSP calculations were used to obtain the crystalline size of the Ag nanocrystals in the Zn_1−x_Ag_x_O compound. As expected, the crystalline size is increased by increasing the Ag amount.

The strain resulting in Hooke’s law, σ = Yε, gives the lattice stress, in which *Y* is Young’s modulus, and *σ* is the stress. However, the validity of this linear relation is restricted to dislocations and a small uniform strain. The SSP method is used to calculate the hexagonal structure; to employ Equation (6), Young’s modulus is found [66]:(6)$$Y_{hkl}=\frac{{\left[h^{2}+\frac{{\left(h+2k\right)}^{2}}{3}+{\left(\frac{al}{c}\right)}^{2}\right]}^{2}}{s_{11}{\left(h^{2}+\frac{{\left(h+2k\right)}^{2}}{3}\right)}^{2}+s_{33}{\left(\frac{al}{c}\right)}^{4}+\left(2s_{13}+s_{44}\right)\left(h^{2}+\frac{{\left(h+2k\right)}^{2}}{3}\right){\left(\frac{al}{c}\right)}^{2}}$$

In the above-mentioned formula, the elastic compliances of ZnO *s*11, *s*13, *s*33, and *s*44 takes the value 7.859 × 10^−12^, −2.205 × 10^−12^, 6.941 × 10^−12^, and 23.57 × 10^−12^ m^2^N^−1^, respectively [67]. The obtained value of *Y* is 360 GPa for the ZnO NPs. Also, using the following equation, the energy density of the lattice is computed:u= σ22Yhkl=ε2Yhkl2

Table 2 presents the results of the SSP and Scherrer methods. It is indicated that by increasing the Ag dopant concentration, the crystallite size of the NPs gained from the SSP and Scherrer techniques decreases. Also, the kinetics of crystal growth is seriously affected by the chemical reactivity of the dopant [68]. In brief, as shown in Table 2, the growth morphology of the NPs is mediated by the effect of the Ag defect.

### 3.2. TEM Analysis

The TEM micrograph studies shown in Figure 7 also indicated that the particle size of the ZnO nanoparticles increased as the Ag amount increased. The average crystallite size of pure ZnO nanoparticles was found to be 30 nm, and the sizes for Ag-doped nanoparticles increase by increasing the doping concentration [69]. These findings were in satisfactory alignment with the XRD results.

### 3.3. FTIR

The characteristic peaks of *κ*-Carrageenan hydrogels (A) and *κ*-Carrageenan hydrogels/Zn_1−x_Ag_x_O (B) x = 0.00, (C) x = 0.03, (D) x = 0.06, and (E) x = 0.09/catechin are present in Figure 8. The figure shows that the peak at 3429 cm^−1^ is because of O–H stretching. The peaks as a result of the C–H stretching vibration of the alkane groups and the S–O of the sulfate esters appeared at 2933 and 1247 cm^−1^, respectively. At 947 cm^−1^, 3,6-anhydro-D-galactose was observed. The peak that appeared at 833 cm^−1^ was because of the galactose-4-sulfate [70,71]. The peaks detected at 702–1094 cm^−1^ were correlated to the ring of benzene (1,2-distributed and 1,3-distributed) in the catechin. The C=O bonds are confirmed at 1665 cm^−1^. The C–O alcohol ingredient appears at 1261 cm^−1^. The peaks that emerged at 1400 cm^−1^ to 1600 cm^−1^ relate to the C=C stretching vibration of the alkane medium and aromatic compound, respectively [72]. The results of the FTIR confirm that all the nanoparticles have made chemical bonds in the network of the hydrogel.

### 3.4. TGA and DSC Analysis

Figure 9A depicts the TGA curve of bare ZnO NPs and Ag-ZnO NPs. The heating process started at 20 °C and, after that, increased up to 1000 °C with a rate change of 5 °C /min. The TGA diagrams show three weight loss regions: the first observation at 20 and 220 °C is related to the water molecule’s evaporation and disintegration of organic compounds [73], and the second part happened at 220 and 380 °C and is attributed to the decomposing of the functional groups. The last weight loss from 380 to 680 °C is associated with the formation of the decomposition of the pyrochlore phases and pyrochlore phases [8,74,75]. The TGA diagram tends to be flat and no less than a weight between 650 and 900 °C, so the NPs are thermally stable above 650 °C.

Figure 9B shows the differential scanning calorimetry (DSC) curve of the Ag-ZnO NPs. A small low-temperature endothermic peak at 80.24 °C is due to the loss of a volatile surfactant molecule adsorbed on the surface of the Ag-ZnO NPs during synthesis conditions. A large high-temperature endothermic peak at 323.73 °C is assigned to the conversion of the Ag-ZnO NPs. A small high-temperature endothermic peak at 683.05 °C is attributed to the conversion of the Ag-ZnO NPs.

### 3.5. Texture

Compression tests were performed to calculate the texture properties of the hydrogels. Table 3 shows the effect of silver nanoparticles on the texture properties of κ-Carrageenan in the hydrogel.

The highest amount of stiffness of the hydrogel structure reached
148.413 ± 4.17 (g) among the samples. The existence of Ag at a low level improved the strength of the hydrogel. However, Ag at higher concentrations showed a lower hardness. This may be associated with the larger porous structure of hydrogel in the existence of Ag in great concentrations. Nevertheless, the lowest amount of stiffness is related to Zn_0.91_Ag_0.09_O, which is 50.319 ± 0.06 (g), and the highest amount of stiffness is associated with Ag at 3%, which is 148.413 ± 4.17 (g). The adhesiveness of the hydrogels had a comparable reaction with the stiffness. The highest amount of adhesiveness was associated with Zn_0.94_Ag_0.06_O, which was 432.086 ± 4.50 (g.s); yet, the lowest amount was presented for Zn_0.91_Ag_0.09_O, with
150.264 ± 2.13. The highest amount of springiness was 7.334 ± 0.62, which was related to Zn_0.94_Ag_0.06_O, and the lowest springiness was related to Zn_0.97_Ag_0.03_O, with 5.645 ± 0.46 (g.s). The adhesion was also affected by the Ag nanoparticles. The lowest rate of adhesion was related to Zn_0.97_Ag_0.03_O, with −1.967 ± 0.19 (g.s), and the highest rate of adhesion was related to Zn_0.91_Ag_0.09_O, with −0.116 ± 0.16 (g.s). Therefore, according to the results of the compression test, the highest rate of springiness and adhesiveness was related to Zn_0.94_Ag_0.06_O, the highest rate of hardness was related to Zn_0.97_Ag_0.03_O, and the highest rate of adhesion was related to Zn_0.91_Ag_0.09_O. It can be concluded that by increasing the concentration of Ag in the hydrogels, the hardness decreases, and the adhesion increases.

### 3.6. FESEM and EDS

Figure 10 shows the FESEM micrographs and the particle size distributions of the bare ZnO NPs and Ag-ZnO NPs. These images indicate that all the synthesized nanoparticles have a spherical morphology. The pure zinc oxide NPs have an average particle size of 30 nm. After we used silver as a dopant, the nanocrystalline gathered and recognized the larger particles. In the EDS spectrum, a strong signal was exhibited from the Zn atom along with a small signal from the O atom and Ag atom, as shown in Figure 10 (I–IV), respectively.

### 3.7. Swelling Ratio and In Vitro Release Studies

Table 4 depicts the drug content of the prepared hydrogels. The polymeric combination with the nanoparticles exhibited good hydrogel-forming properties. The drug content in the hydrogels range from 97.71 ± 1.02% to 98.59 ± 2.01%. The results indicated that the method selected for the preparation of the hydrogels was capable of producing hydrogels with a high drug content.

The swelling ratio of the hydrogel quantitatively performs the looseness of the inner structure of the hydrogel network, which is the indicator of the hydrogel network structure. The higher rate presents a looser inner structure of the hydrogel. Figure 11A shows that the κ-Carrageenan hydrogels reach the swelling equilibrium through 180 min and reveals that the water absorption, step by step, increases with a growth in the silver content. It reveals an increase in the swelling ratio of the κ-Carrageenan hydrogels by increasing the content of the Ag ions during the early stage, and it takes an extended period of time to get to the swelling equilibrium. The slope of the swelling curve for the Ag-free sample (κ-Carrageenan/ZnO/NaCMC/catechin) hydrogel is the smallest, and the time mandatory to reach the equilibrium is less than 100 min. The slope of the swelling curve for the hydrogels with a higher concentration of Ag ions increased with the growth of the Ag dosage. This can be attributed to the following reasons: With the growth of the Ag concentration, the number of silver ions in the hydrogel structure grows, thus enhancing the osmotic pressure of the inner structure and increasing the relaxation of the structure chain. The enhancement in the water sorption capacity is conducive to higher swelling. The time-dependent in vitro release of catechin from the hydrogels over 5 h from all the prepared hydrogel formulations is shown in Figure 11B. The differences in the release profile of the ZnAgO hydrogels after 5 h were evaluated using a graph plotting the absorbance vs. the concentration of the known solutions. The control hydrogel without any silver nanoparticle showed the lowest release rate (32 ± 2.34%) within 300 min, which might be due to the less strength of the hydrogel compared with others. The resulted release patterns of the Ag-loaded nanoparticles in the κ-Carrageenan hydrogels might be attributed to the stable binding between the Ag nanoparticles and the κ-Carrageenan hydrogel matrix. Therefore, the highest release is related to the highest concentration of Ag nanoparticles (84 ± 2.18%) within 300 min that are coated with ZnO nanoparticles (Figure 11).

## 4. Conclusions

ZA-NPs were synthesized by the gelatin stabilized sol-gel technique with various concentrations of Ag (x = 0.0, 0.03, 0.06, and 0.09), and the obtained gels were thermally treated at 700 °C for 2 h to gain fine powders. An XRD structural analysis indicated that Ag and ZnO nanocrystals grow independently and, therefore, both cubic and hexagonal diffraction peaks were observed in the pattern. The SSP analysis of the XRD data showed that the crystalline size of the silver and ZnO nanocrystals increased when the concentration of Ag increased in the composite. The particle sizes were measured from the TEM micrograph and were obtained to be between 27 and 36 nm. The texture analysis showed that by the growth of the concentration of Ag in the hydrogels, the hardness decreases, and the adhesion increases. With an increase in the Ag concentration, the relaxation of the network chain was increased. The highest drug content and release are related to the highest concentration of Ag nanoparticles that are coated with ZnO nanoparticles. As future prospects, it can be suggested to modify and improve the matrix by some crosslinking methods in polymerization and use this compound as a suitable and antibacterial food packaging. The catechin also has an antioxidant effect that can be useful and beneficial for this reason.

## Figures and Tables

**Figure 1 materials-15-05536-f001:**
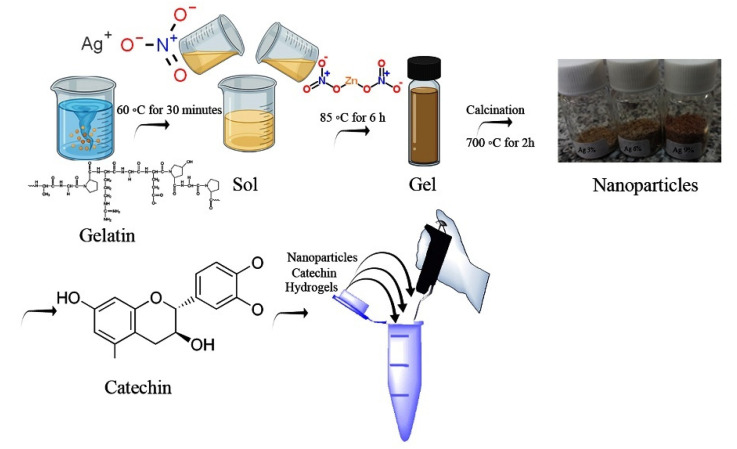
Schematic diagram of nanoparticles’ synthesis.

**Figure 2 materials-15-05536-f002:**
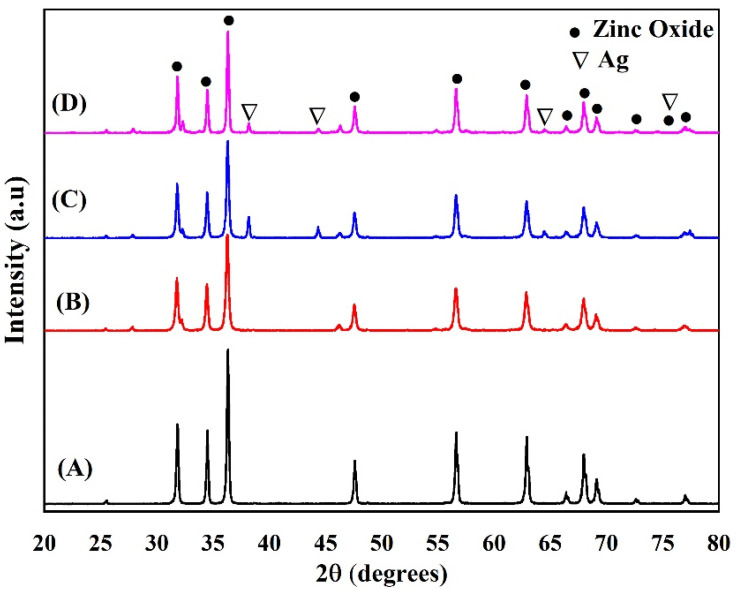
XRD patterns of Zn_1−x_Ag_x_O. (A) x = 0.0, (B) x = 0.03, (C) x = 0.06, and (D) x = 0.09.

**Figure 3 materials-15-05536-f003:**
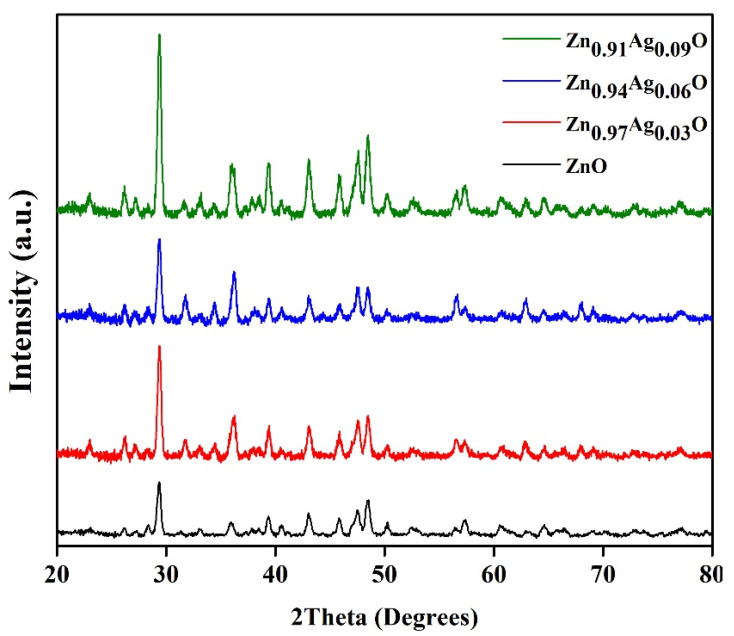
XRD patterns of *κ*-Carrageenan/ZA-NPs/NaCMC/catechin.

**Figure 4 materials-15-05536-f004:**
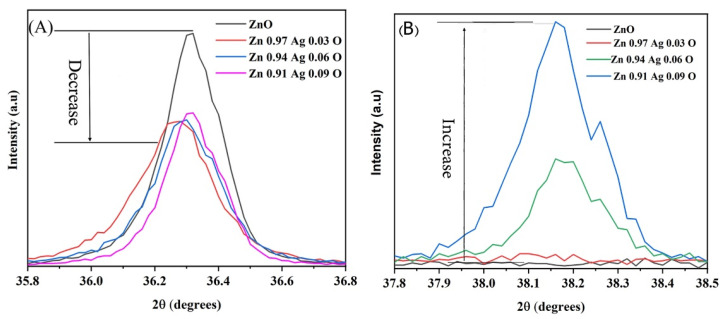
The XRD peak related to (**A**) the ZnO hexagonal and (**B**) Ag cubic structures.

**Figure 5 materials-15-05536-f005:**
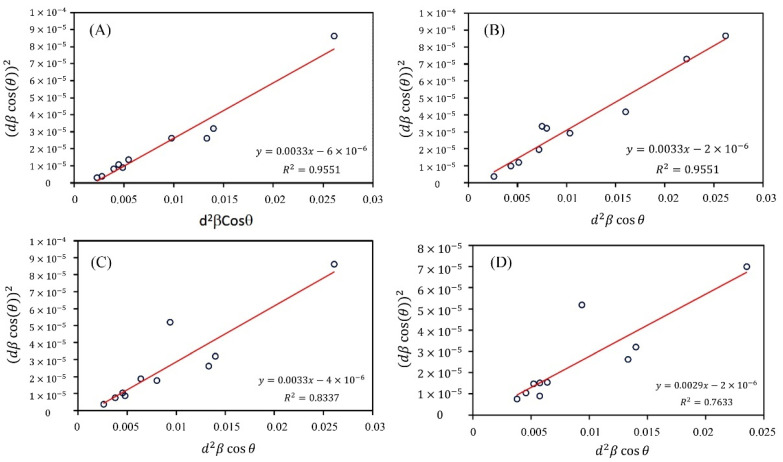
SSP plot for the ZNO hexagonal diffraction peaks, Zn_1−x_Ag_x_O (**A**) x = 0.0, (**B**) x = 0.03, (**C**) x = 0.06 and (**D**) x = 0.09.

**Figure 6 materials-15-05536-f006:**
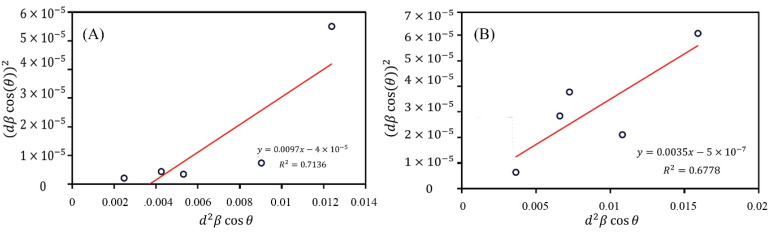
SSP plot for the Ag diffraction peaks in Zn_1−x_Ag_x_O. (**A**) x = 0.06 and (**B**) x = 0.09.

**Figure 7 materials-15-05536-f007:**
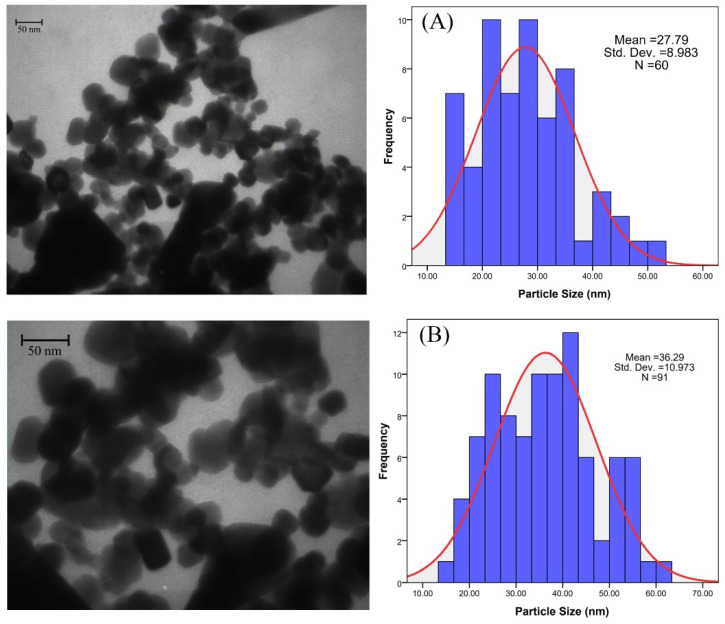
The TEM results of (**A**) the ZnO-NPs. (**B**) Zn_0.97_Ag_0.03_O.

**Figure 8 materials-15-05536-f008:**
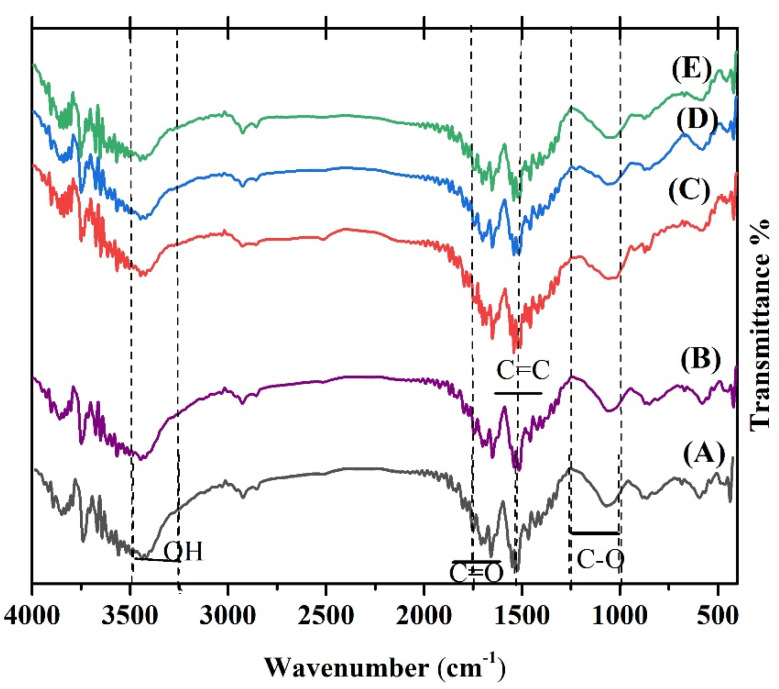
FTIR spectra of κ-Carrageenan hydrogels (A) and *κ*-Carrageenan hydrogels/Zn_1-x_Ag_x_O. (B) x = 0.00, (C) x = 0.03, (D) x = 0.06, and (E) x = 0.09/catechin.

**Figure 9 materials-15-05536-f009:**
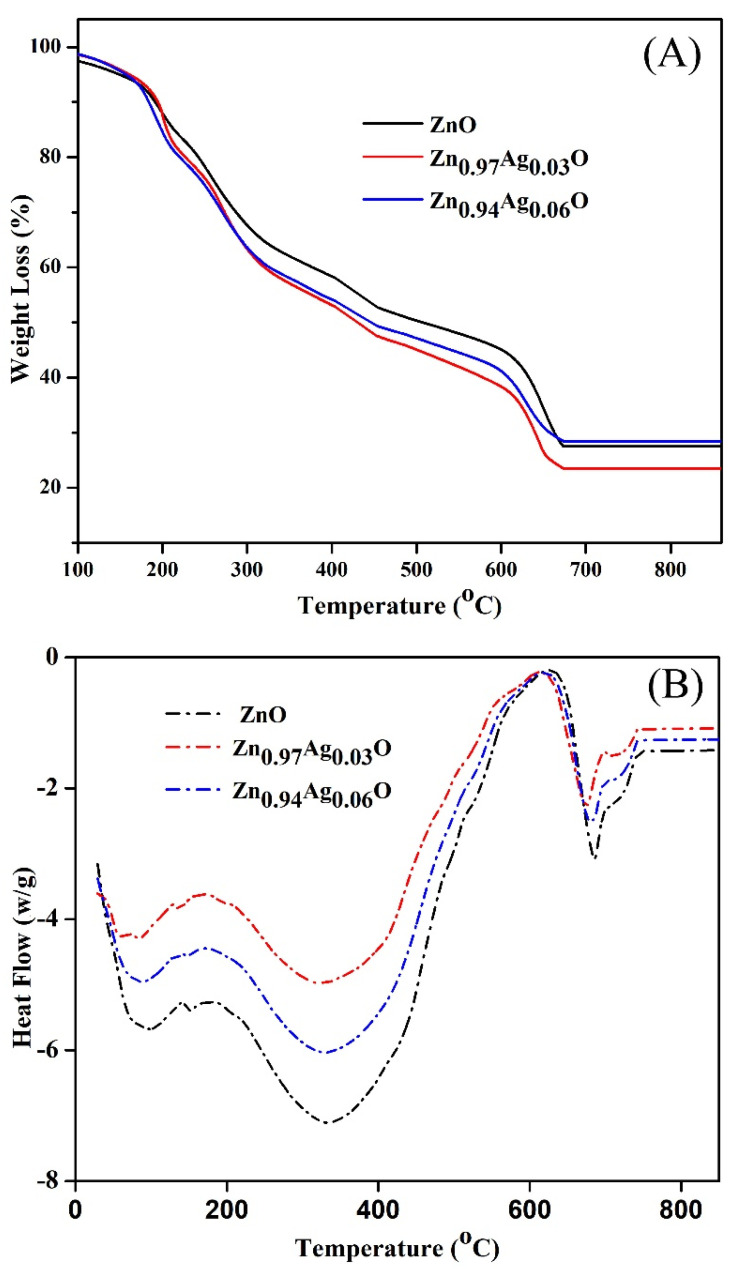
(**A**)Thermal analysis result of the synthesized Ag-ZnO NPs and (**B**) the differential scanning calorimetry (DSC) curve of the Ag-ZnO NPs.

**Figure 10 materials-15-05536-f010:**
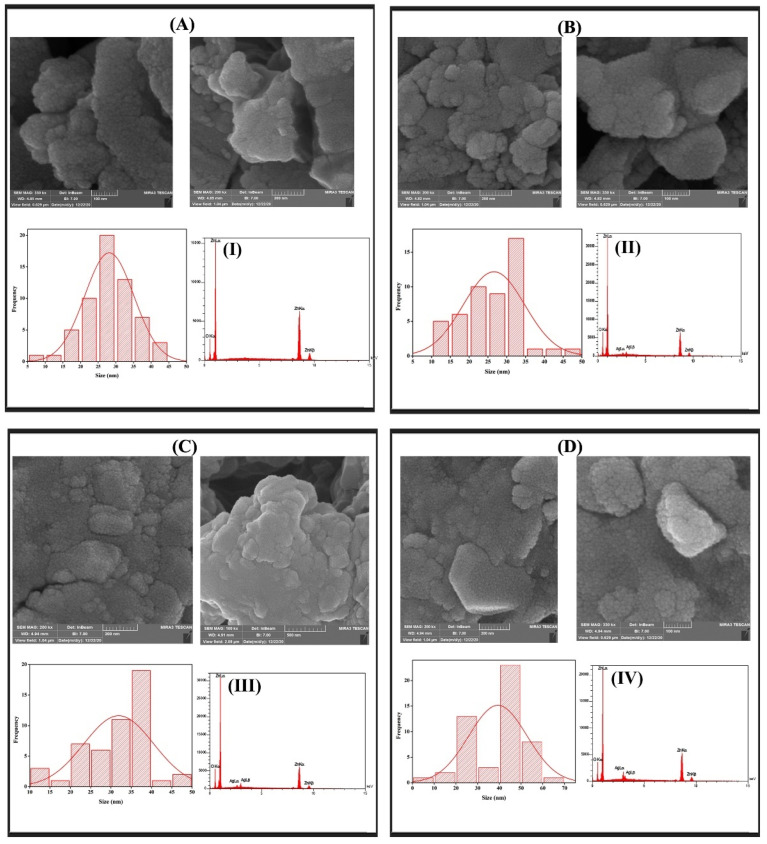
The FESEM images and EDS spectrum of the prepared Zn_1−x_Ag_x_O. (**A**) x = 0.0, (**B**) x = 0.03, (**C**) x = 0.06, and (**D**) x = 0.09.

**Figure 11 materials-15-05536-f011:**
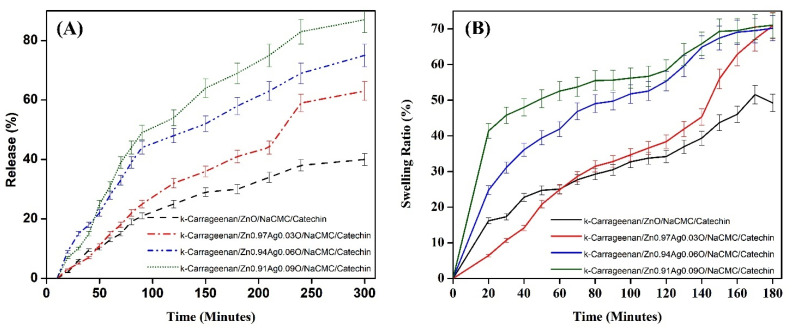
(**A**) Time-dependent in vitro release of catechin from (**B**) the swelling equilibrium of the hydrogels and (**B**) the time-dependent in vitro release of catechin from the hydrogels.

**Table 1 materials-15-05536-t001:** The numerous combinations of ZnO and the Ag-doped XRD peak, lattice parameters, and their crystal structure and volume.

Compound	2θ ± 0.01	*hkl*	*dhk l* (nm) ± 0.0005	Structure	Lattice Parameter (nm) ± 0.0005	V(Å^3^) ± 0.0002
ZnONPs	31.8291 34.4884	(100) (002)	0.2811 0.2600	Hexagonal	a = 0.32465 c = 0.520122	0.0474737
Zn_0.97_Ag_0.03_O	31.7693 34.4491	(100) (002)	0.2816 0.2603	Hexagonal	a = 0.325246 c = 0.520696	0.0477007
Zn_0.94_Ag_0.06_O	31.8083 34.4688	(100) (002)	0.2813 0.2602	Hexagonal	a = 0.324857 c = 0.520408	0.0475603
Zn_0.91_Ag_0.09_O	31.8389 34.4876	(100) (002)	0.2810 0.2600	Hexagonal	a = 0.324553 c = 0.520132	0.0474463

**Table 2 materials-15-05536-t002:** Ag concentration-dependent elastic constants and the NPs’ sizes.

Compound	Scherrer	Size Strain Plot				
	D (nm)	D (nm)	$\;\varepsilon \times 10^{-3}$	$Y\times 10^{9}$	$\;\sigma \times 10^{7}$	$u\times 10^{4}$
ZnO-NPs	42	28	3.46	360	12.48	21.61
Zn_0.97_Ag_0.03_O	45	35	1.55	360	55.81	43.23
Zn_0.94_Ag_0.06_O	47	39	1.26	360	45.57	28.82
Zn_0.91_Ag_0.09_O	53	48	1.55	360	55.81	43.23

**Table 3 materials-15-05536-t003:** Effect of Ag nanoparticles on the textural characterization of κ-Carrageenan in the hydrogel.

Sample	Hardness (g)	Adhesiveness	Springiness (mm)	Adhesion (g.s)
Zn O	127.476 ± 3.40	328.787 ± 3.40	6.299 ± 0.52	−1.619 ± 0.12
Zn_0.97_Ag_0.03_O	148.413 ± 4.17	275.727 ± 2.10	5.645 ± 0.46	−1.967 ± 0.19
Zn_0.94_Ag_0.06_O	133.838 ± 3.15	432.086 ± 4.50	7.334 ± 0.62	−1.504 ± 0.15
Zn_0.91_Ag_0.09_O	50.319 ± 0.06	150.264 ± 2.13	6.999 ± 0.73	−0.116 ± 0.16

**Table 4 materials-15-05536-t004:** Various physicochemical properties of the prepared hydrogels.

Sample	Drug Content (%)
ZnO-NPs	97.71 ± 1.02%
Zn_0.97_Ag_0.03_O	98.59 ± 2.11%
Zn_0.94_Ag_0.06_O	98.41 ± 1.14%
Zn_0.91_Ag_0.09_O	97.91 ± 1.32%

## Data Availability

Not applicable.

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
