# Peer review of "Green Facile Synthesis of Silver-Doped Zinc Oxide Nanoparticles and Evaluation of Their Effect on Drug Release"

_materials, 2022, doi:10.3390/ma15165536_

Round 1
Reviewer 1 Report
Dear Editor,
I have read the manuscript entitled: “Green facile synthesis of silver-doped zinc oxide nanoparticles 2 and evaluate their effect for drug release ” and I would like to address following suggestions to the authors:
Line 16: “sizes these nanoparticles ” should be “sizes of these nanoparticles”
Lines 44-47: Methods “combustion syn-45 thesis, pyrolysis” appear mentioned in two places in the same sentence
Line 45: reserve micelle [30],” should be “reserve micelle [30],
line 47: plasma syn-thesis [38],” should be “plasma synthesis [38],
Line 55: mediated defect ,” should be “mediated defects
Throughout the manuscript “dopped” should be “doped”. Please check and correct.
Line 88: this liquid” should be “ This liquid
Line 148: Where is the content of Table 1?
Line 256: last wight loss” should be “last weight loss
Line 291: Figure 9 show” should be “ Figure 9 shows
Line 293: Please rewrite this:"The pure znic oxide hase a.. “
Line 337: The particles sizes ” should be “The particle sizes
Author Response
1-line 16
2-Done
3-line 47
4-line 48
5-line 56
6-lines 42-43-68-171-251
7-line 92
8-Lines: 171-172 are the content of Table 1.
9-line 282
10-line 326
11-The pure zinc oxide NPs has an average particle size of about 30 nm. Line:328
12-line 386

Reviewer 2 Report
Review
The topic seems interesting, but there are some issues that need clarification or correction.
I would suggest the following questions and comments to be taken into consideration:
1) Revise the manuscript, the English is not completely satisfactory. All typos should also be corrected.
2) Abstract and conclusion should contain more quantitative information.
3) Please add some lines to indicate the novelty of your study, compare the results with that of the literature and emphasize the novelty of this study. My main concern is that there are similar manuscripts related to the topic (e.g. Inorganic Chemistry Communications, 2021, 131: 108762.) Try to describe in detail the novelty of this manuscript. Some attention should be paid in the introduction to cite newer results.
4) A more detailed description of the measurements used is required, e.g. What was the scanning rate and range for XRD measurements? How were the structural and textural properties (e.g. the Zn and Ag content) of the samples determined? What about the particles stability?
5) The calcination carried out at 700 °C for 2 h. What are the reasons that these high energy consuming methods was chosen in this green paper? How were these energies taken into account when mentioning environmental friendly-green method?
6) It would be nice to see some basic biomedical test on this drug delivery system.
7) In the conclusion, the performance findings of the research should have been summarized the innovations and future scope of the work should be highlighted more.
8) Figure 5 is hard to read, please improve the understanding and readability.
Author Response
1- Done
2- Lines:19-21
3- Thank you for your good comment. line 67-74
Each element in the periodic table has different properties. Zinc and cerium are different from each other.
Zinc:
Atomic number (Z) 30
Group: group 12
Period: period 4
Electron configuration: [Ar] 3d10 4s2
Crystal structure: hexagonal close-packed (hcp)
Cerium:
Atomic number (Z) 58
Period: Period 6
Electron configuration: [Xe] 4f1 5d1 6s2
Crystal structure: face-centered cubic (fcc)
The article mentioned by the respected referee mentions the element cerium. Its structure and morphology are entirely different. The results of XRD and TGA confirm this article.
4- Thank you for your good comment. Lines:127-128
(2θ=20⁓80°, rate of 5 deg/min)
As shown in figure 1, Zn and Ag content with ● and Δ
Zn: 31.84°, 34.44°, 36.37°, 47.45°, 56.76°, 62.93°, 66.50°,68.00°,69.23°, 72.65°, 77.04°
Ag: 38.28°, 44.44°, 64.57°
Zn and Ag: 74.58°
One of the best factors to evaluate the stability of the nanoparticles is particle size [1]. To all of steps, particle size is stable because we used gelatin as a stabilizer in synthesis process [2, 3]. Lines: 70-72
- Singh, R.P., K. Sharma, and K. Mausam, Dispersion and stability of metal oxide nanoparticles in aqueous suspension: A review. Materials Today: Proceedings, 2020. 26: p. 2021-2025.
- Bashir, M., et al., Biodegradation of gelatin stabilized tetragonal zirconia synthesized by microwave assisted sol-gel method. Journal of the Mechanical Behavior of Biomedical Materials, 2022. 127: p. 105070.
- Kazemi, M., et al., Evaluation of antifungal and photocatalytic activities of gelatin-stabilized selenium oxide nanoparticles. Journal of Inorganic and Organometallic Polymers and Materials, 2020. 30(8): p. 3036-3044.
5-
Thank you for your good comment.
In order to prepare nanoparticles via sol gel method, we have 2 main steps. First, Chemicals were implemented to obtain sol-gel. A viscous and transparent gel turned into acquired because of constantly stirring the solution at 85 ◦C for 6 h.
Second, the resulting gel was rubbed for the calcination process on the inner wall of a small amount of alumina crucible and put into a furnace at 700 °C. The rate of heating was 5°C/min, and the duration of heating time was 2 hours.
To calcinate we need this temperature (700 °C) to produce nanoparticles via sol-gel method. This amount is different to each material.
Many methods are available to produced nanoparticles via green synthesis. One of them is sol-gel method because of the materials.In this work, Ag-doped ZnO NPs were synthesized by so-gel method using gelatin as a natural polymerization agent. This method is green method because of use of gelatin[4-9].
- Darroudi, M., et al., Time-dependent effect in green synthesis of silver nanoparticles. International journal of nanomedicine, 2011. 6: p. 677.
- Sabbagh, F., et al., Effect of zinc content on structural, functional, morphological, and thermal properties of kappa-carrageenan/NaCMC nanocomposites. Polymer Testing, 2021. 93: p. 106922.
- Maleki, P., et al., Green facile synthesis of silver-doped cerium oxide nanoparticles and investigation of their cytotoxicity and antibacterial activity. Inorganic Chemistry Communications, 2021. 131: p. 108762.
- Sabbagh, F., et al., Green synthesis of Mg0. 99 Zn0. 01O nanoparticles for the fabrication of κ-Carrageenan/NaCMC hydrogel in order to deliver catechin. Polymers, 2020. 12(4): p. 861.
8. Suarasan, S., et al., One-pot, green synthesis of gold nanoparticles by gelatin and investigation of their
biological effects on Osteoblast cells. Colloids and Surfaces B: Biointerfaces, 2015. 132: p. 122-131.
9. Zak, A.K., et al., XPS and UV–vis studies of Ga-doped zinc oxide nanoparticles synthesized by gelatin based sol-gel approach. Ceramics International, 2016. 42(12): p. 13605-13611.
6- Thank you for your good comment. Lines: 339-345
The drug content data of the hydrogels are provided in Table 4.
7- Thank you for your good comment. Lines:391-394
As future prospects, it can be suggested to modify and improve the matrix by some crosslinking methods in polymerization, and use this compound as a suitable and antibacterial food packaging. The catechin also has an antioxidant effect that can be useful and beneficial for this reason.
8- Done

Reviewer 3 Report
1. Line 118 In the paragraph is XRD all devices are described except FESEM and TGA/DSC. Need to add.
2. Line 158. Under what conditions and rates of heating and cooling did the heat treatment of nanoparticles take place?
3. Line 228. Judging by the figure, on the contrary, with an increase in the silver concentration, the particle size decreases, since pure ZnO = 36.3 nm, and Zn0.97Ag0.03 = 27.8 nm. It is necessary to clarify the result and expand the discussion.
4. Line 248. On what basis did you decide that all nanoparticles form bonds with the polymer matrix? The graphs you provided are identical and contain nanoparticles; for comparison, a pure polymer control plot should be provided.
5. Line 251 Based on thermogravimetry, it is impossible to evaluate and recognize processes occurring at different temperatures. In order to be able to reason, it is necessary to provide a DSC graph; after that, processes can be analyzed. In general, the significance of this graph is not very clear, since at these silver concentrations, phase transformations occur at no less than 800°C.
6. I would like to see a chemical analysis of these nanoparticles (XPS or EDS) and add the results to the discussion.
7. For each research method, it is necessary to expand the discussion and describe by chemical reactions how nucleation and micellization occurs, as well as polymer degradation and other reactions during high-temperature processing.
Author Response
1- Done. Lines: 133-135
Field emission scanning electron microscopy (FESEM) (MIRA III, TESCAN, Czech), Thermal stability was investigated via TGA Thermogravimetric Analysis (TGA) (TA, Q600, USA).
2- Thank you for your good comment. Lines: 96-97
According to the reference that we have used in the methodology, there is no information about the heating rate and in this study, the samples were put into a furnace at 700 °C. The rate of heating was 5°C/min.
3- Thank you for your good comment. Lines: 250-252
The results in the figure were shown in incorrect order by some errors and it is corrected now.
ZnO nanoparticles increased by the Ag amount increases.
Corrected: Figure 6. TEM results of (A) ZnO-NPs. (B) Zn 0.97 Ag 0.03 O.
For more discation:
Average crystallite size of pure ZnO nanoparticles was found to be 30 nm and the sizes for Ag-doped nanoparticles increase by increasing doping concentration [71].
[71] Iqbal, T., et al., Simple synthesis of Ag-doped CdS nanostructure material with excellent properties. Applied Nanoscience, 2020. 10(1): p. 23-28
4- Thank you for your good comment. Lines: 259-363
A pure polymer control plot has been added to the graph.
5- Thank you for your good comment. Lines: 286-294
Figure 9 (B) shows the Differential Scanning Calorimetry (DSC) curve of Ag-ZnO NPs. A small low temperature endothermic peak at 80.24oC is due to the loss of volatile surfactant molecules adsorbed on the surface of Ag-ZnO NPs during synthesis conditions. A large high-temperature endothermic peak at 323.73ºC is assigned for the conversion of Ag-ZnO NPs. A small high-temperature endothermic peak at 683.05 ºC was attributed to the conversion of Ag-ZnO NPs.
6- Thank you for your good comment. Lines: 330-333
EDS has been added to figure 9 (I, II, III, IV)
In the EDS spectrum, a strong signal was exhibited from the Zn atom along with a small signal from the O atom and Ag atom as shown in Fig. 9 (I) and (II, III, and IV respectively.
Figure 10. The FESEM images and EDS spectrum of the prepared Zn1-xAgxO (A) x=0.0, (B) x=0.03, (C) x=0.06 and (D) x=0.09.
7- Thank you for your good comment.Lines:98-100
Figure 1 has been provided.

Round 2
Reviewer 2 Report
The authors answered all the questions, the manuscript has been sufficiently improved to warrant publication in Materials.
Reviewer 3 Report
Accepted.